# Classification of large ornithopod dinosaur footprints using Xception transfer learning

**Yeoncheol Ha**[1], **Seung-Sep Kim**[1,2]*

1 Department of Astronomy, Space Science and Geology, Chungnam National University, Daejeon, Korea,
2 Department of Geological Sciences, Chungnam National University, Daejeon, Korea

* seungsep@cnu.ac.kr

**Data Availability Statement:** The source code and training and test datasets used for this study is available from https://zenodo.org/doi/10.5281/zenodo.10056902. The final trained model is

## Abstract

Large ornithopod dinosaur footprints have been confirmed on all continents except Antarctica since the 19[th] century. However, oversplitting problems in ichnotaxa have historically been observed in these footprints. To address these issues and distinguish between validated ichnotaxa, this study employed convolutional neural network-based Xception transfer learning to automatically classify ornithopod dinosaur tracks. The machine learning model was trained for 162 epochs (i.e., the number of full cycles of all training data through the model) using 274 data images, excluding horizontally flipped images. The trained model accuracy was 96.36%, and the validation accuracy was 92.59%. We demonstrate the performance of the machine learning model using footprint illustrations that are not included in the training dataset. These results show that the machine learning model developed in this study can properly classify footprint illustration data for large ornithopod dinosaurs. However, the quality of footprint illustration data (or images) inherently affects the performance of our machine learning model, which performs better on well-preserved footprints. In addition, because the developed machine-learning model is a typical supervised learning model, it is not possible to introduce a new label or class. Although this study used illustrations rather than photos or 3D data, it is the first application of machine-learning techniques at the academic level for verifying the ichnotaxonic assignments of large ornithopod dinosaur footprints. Furthermore, the machine learning model will likely aid researchers to classify the large ornithopod dinosaur footprint ichnotaxa, thereby safeguarding against the oversplitting problem.

## Introduction

Dinosaur footprints provide paleoecological information about pace, stride, speed, gait behavioral characteristics that are difficult to determine from body fossils [1]. Historically, large ornithopod dinosaur footprints were associated with *Iguanodontidae* [2], and have been confirmed from the 19[th] century to the present on all continents except Antarctica [3]. However, it is important to note that different species can produce similar tracks, and even a single track-maker species can generate tracks with variable morphologies. Furthermore, oversplitting issues (where species or other taxonomic groups are divided into too many smaller groups based on minor differences) of ichnotaxa in large ornithopod dinosaur footprints have been

downloadable from https://zenodo.org/doi/10.5281/zenodo.10056870.

**Funding:** This study was supported by the National Research Foundation of Korea (NRF-2021R1A4A5026233 and NRF-2021R1A2C1012030), awarded to S.-S.K. The funder had no role in study design, data collection and analysis, decision to publish, or preparation of the manuscript. There was no additional external funding received for this study.

**Competing interests:** The authors have declared that no competing interests exist.

recognized [2–5] due to various factors. For instance, the definition of ichnotaxa is sometimes based on poorly preserved samples or lacks diagnostic features [6], inadequate diagnostics [2], or temporal or geographic criteria [3, 7]. In addition, marking and confirming the outline of dinosaur footprints with a material such as chalk [8–11] can be affected by the researchers' subjectivity and experience and provide no depth information of footprints.

To circumvent such issues, 3D techniques such as photogrammetry and light detection and ranging (LiDAR) have recently been introduced for digitizing and recording dinosaur footprints [10–26]. Furthermore, Lallensack [27] produced a program to create objective track outlines from 3D footprints in R environment. Although these studies can effectively alleviate the significance of subjectively recording and digitizing dinosaur footprints, the oversplitting problem associated with the classification of large ornithopod ichnotaxa has not yet been fully addressed in terms of automation.

For example, the scientific names of the large ornithopod dinosaur ichnotaxa have been reviewed and renamed. Based on their overall morphology [2], large ornithopod footprints have been classified as *Amblydactylus gethingi* [28], *Caririchnium* [29], *Iguanodontipus* [6], and *Hadrosauropodus* [2]. The Early Cretaceous large ornithopod dinosaur footprints were classified as *Iguanodontipus* (footprints of *Iguanodon*), *Amblydactylus* (strictly bipedal), and *Caririchnium* (quadrupedal) [30]. However, one study argued that all large ornithopod dinosaur footprints should be classified as *Amblydactylus* (*Iguanodontipus*) or *Caririchnium* (= *Hadrosauropodus*) [5], although the study failed to differentiate between *Hadrosauropodus* [2] and *Caririchnium*. Another study classified ornithopod dinosaur footprints into *Iguanodontipus*, *Caririchnium*, and *Hadrosauropodus* based on their biochronological and morphological parameters, including footprint size [4]. However, a recent study defined only *Iguanodontipus*, *Caririchnium*, and *Hadrosauropodus* among 34 large ornithopod dinosaur footprints as valid ornithopod ichnotaxa, based on the size of the pes, heel, and digit impressions, which exhibit mesaxony, tridactyl, and subsymmetrical characteristics [3]. Fig 1 summarizes the changes in classification schemes over the last two decades. In this study, we developed a machine learning classification method for large ornithopod dinosaur footprints (Table 1) based on the classification criteria of Díaz-Martínez et al. [3].

Notably, the aforementioned classification schemes require follow-up studies to manually identify and classify all ichnotaxon diagnostics. Thus, ichnotaxon misnaming problems can still occur because of subjective measures of outlining and experience. Therefore, this study attempted to automatically classify ornithopod dinosaur tracks using convolutional neural networks to circumvent these problems and to distinguish between validated ichnotaxa (i.e., *Caririchnium*, *Hadrosauropodus*, and *Iguanodontipus*) [3].

## Methods

### Convolution neural network and transfer learning

We developed a machine learning model for classifying the outline images of large ornithopod dinosaur footprints using a convolutional neural network (CNN). Since the first appearance of CNN for image classification [31], many researchers have produced various CNN models, such as LeNet [32], VCG-Net [33], GoogleNet [34], and ResNet [35]. Such CNNs generally comprise multiple layers, and each layer performs specific functions in the network as follows [36]. A convolution layer extracts features from the input image and produces a 2D activation map that exhibits features detected at a given location in the input image [31]. A nonlinearity layer introduces nonlinearities into the network. The pooling layer reduces the number of parameters in the network by reducing the spatial resolution of the feature maps. Finally, a

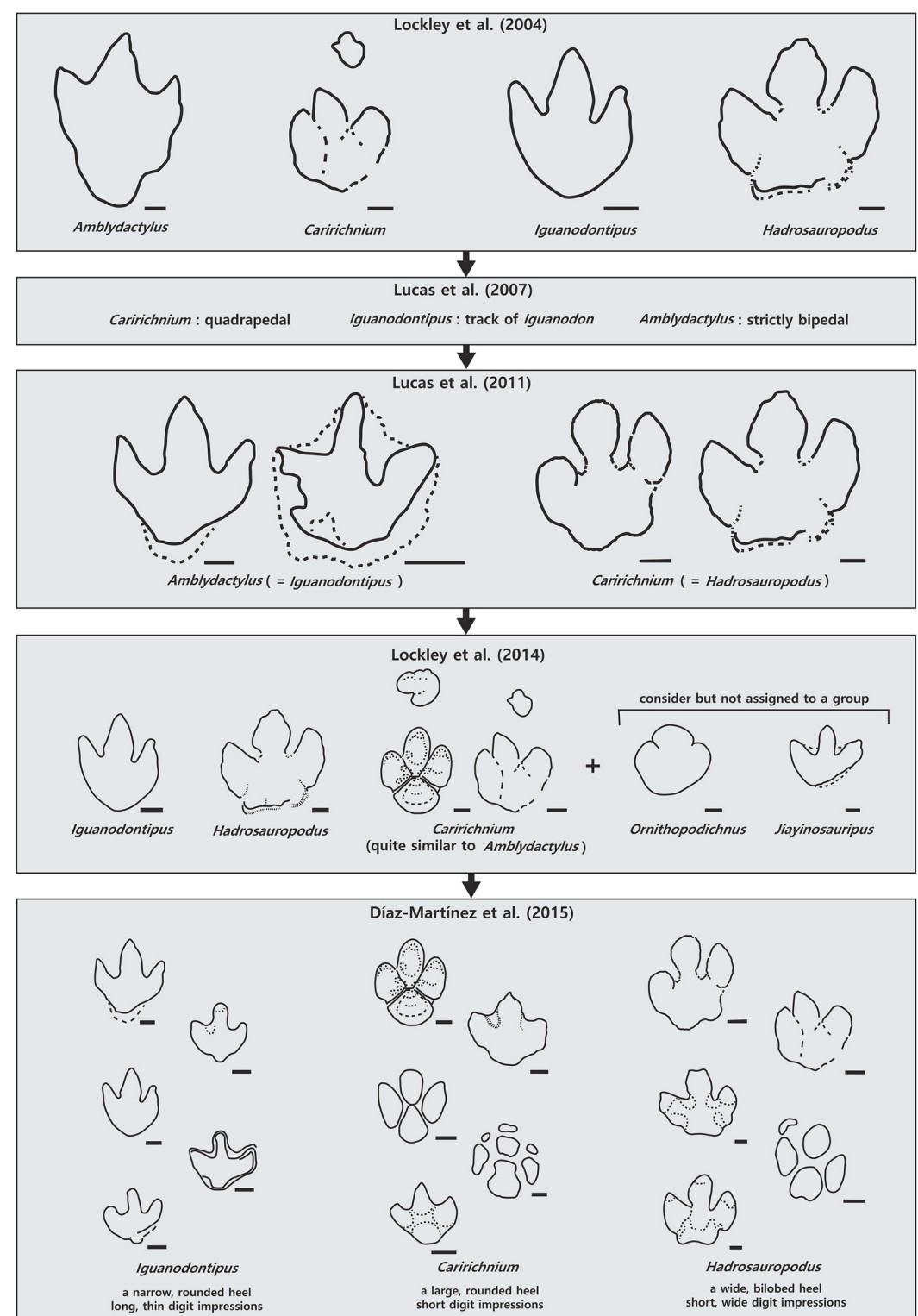

**Fig 1. Changes in large ornithopod ichnotaxa classification since 2004.** The scale bar is 10 cm.

**Table 1. Large ornithopod footprint classification validated by Díaz-Martínez et al. [3].**

| Previous classification | Díaz-Martínez et al. [3] |
|---|---|
| *Amblydactylus kortmeyeri* | *Caririchnium kortmeyeri* |
| *Caririchnium magnificum* | *Caririchnium magnificum* |
| *Caririchnium leonardii* | *Hadrosauropodus leonardii* |
| *Caririchnium lotus* | *Caririchnium lotus* |
| *Caririchnium kyoungsookimi* | *Hadrosauropodus kyoungsookimi* |
| *Hadrosauropodus langstoni* | *Hadrosauropodus langstoni* |
| *Iguanodontipus burreyi* | *Iguanodontipus burreyi* |
| *Iguanodontipus billsarjeanti* | *Caririchnium billsarjeanti* |

fully connected layer is used to generate the final prediction from the feature maps produced by the previous layers.

Among the different variants of CNNs, we utilized Xception [37] to analyze footprint outlines of large ornithopod dinosaurs. Typical convolutions in CNNs operate on each feature map (i.e., channel) independently of the input (i.e., depthwise convolution) or the entire feature map (i.e., pointwise convolution). In Xception, as an alternative to these convolutions, a depth-wise separable convolution is introduced for computational efficiency, which operates on each channel first with depth-wise convolution and then applies point-wise convolution to increase the number of channels in the output. Xception has 14 modules and 36 convolution layers (Fig 2) and is a lightweight model of Inception V3 [37]. However, Xception differs from Inception v3 because it prevents the loss of information by performing depthwise separable convolution and removing nonlinear functions from the operation [37].

Although Xception is known for its good performance and computational efficiency [37], our new machine learning model is inherently constrained by the small number of available ornithopod dinosaur footprint illustrations. To circumvent this problem, we employed transfer learning to improve the performance of the new model [38] by utilizing Xception as a feature extractor and weights trained using ImageNET. ImageNET weights were employed to enhance the learning performance and accuracy of the machine learning model. Instead of the original fully connected layer, our model incorporated a classification layer into a network consisting of two dense layers, one with 64 and 32 nodes weighted by the Rectified Linear Unit (ReLU) function and the other with three nodes weighted by the softmax function (Fig 2). The answer labels were *Caririchnium*, *Hadrosauropodus*, and *Iguanodontipus* (Fig 1).

To avoid any overfitting errors owing to the small training datasets, we also applied an L2 regulation of 0.001 for weight attenuation and a dropout layer to temporarily remove random nodes from the network [39]. As an optimizer, Adam (Adaptive Moment Estimation) [40] was implemented to the transfer learning with the learning rate of $1 \times 10^{-6}$.

## Data collection and preprocessing

Following the nomenclature proposed by Díaz-Martínez et al. [3], we collected 274 dinosaur footprint illustrations from the literature: 111 *Caririchnium* [8–10, 21, 24, 29, 41–44], 36 *Hadrosauropodus* [2, 14, 41, 45–50], and 127 *Iguanodontipus* [13, 51–53]. The geographical distributions of these footprints are listed in Table 2. The collected illustrations (i.e., images) were redrawn using Adobe Illustrator to precondition the image data. For example, unnecessary internal structures and manus data were removed (Fig 3) because the ichnotaxonomy of the large ornithopod dinosaur footprints was revised based only on the contours of the hind

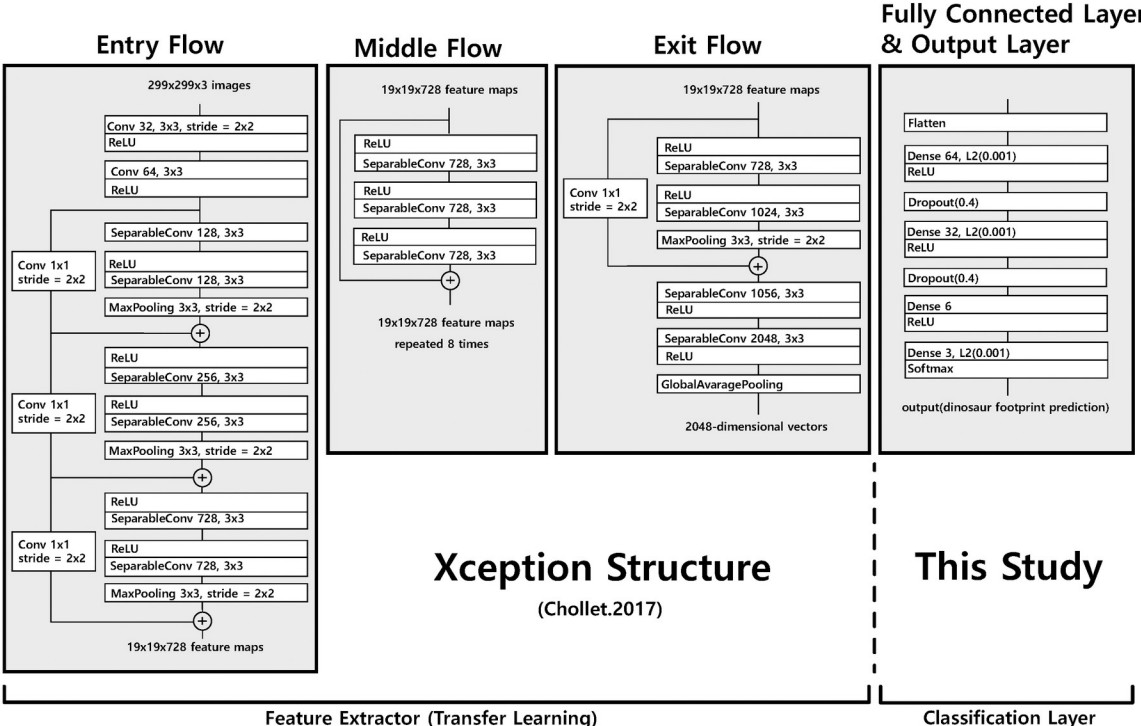

**Fig 2. Schematic architecture of the machine learning model developed for large ornithopod ichnotaxa classification in this study.**

feet [3]. In this study, we used only pes outlines. In addition, all image data in the training set were flipped horizontally and stored individually, in order to generate opposite footprints for each track.

The machine learning model developed in this study is inherently subject to a relatively small amount of dinosaur footprint data. To prevent overfitting, we utilized the ImageData-Generator for random transformation and normalization of the given image datasets, which is a class in the Keras deep learning library [54]. Data augmentation with the ImageDataGenerator was implemented by rescaling, rotating ($< 10°$), shifting width ($< 10\%$), height ($< 10\%$), and zooming ($< 20\%$), which operated randomly on the preconditioned image dataset. Shearing the input images in the ImageDataGenerator was not used in this step to preserve the characteristics of the dinosaur footprints (e.g., bilobed heel).

To classify the outline images of the large ornithopod dinosaur tracks effectively, the input images for the machine learning model developed in this study were prepared as follows: Foot-print images were prepared to show single black-lined pes with no internal structures or

**Table 2. Geographic distribution of the large ornithopod dinosaur track data used in this study.**

|  | *Caririchnium* | *Hadrosauropodus* | *Iguanodontipus* |
|---|---|---|---|
| North America | 3 | 16 | x |
| South America | 4 | 1 | x |
| Asia | 101 | 6 | x |
| Europe | 3 | 13 | 127 |

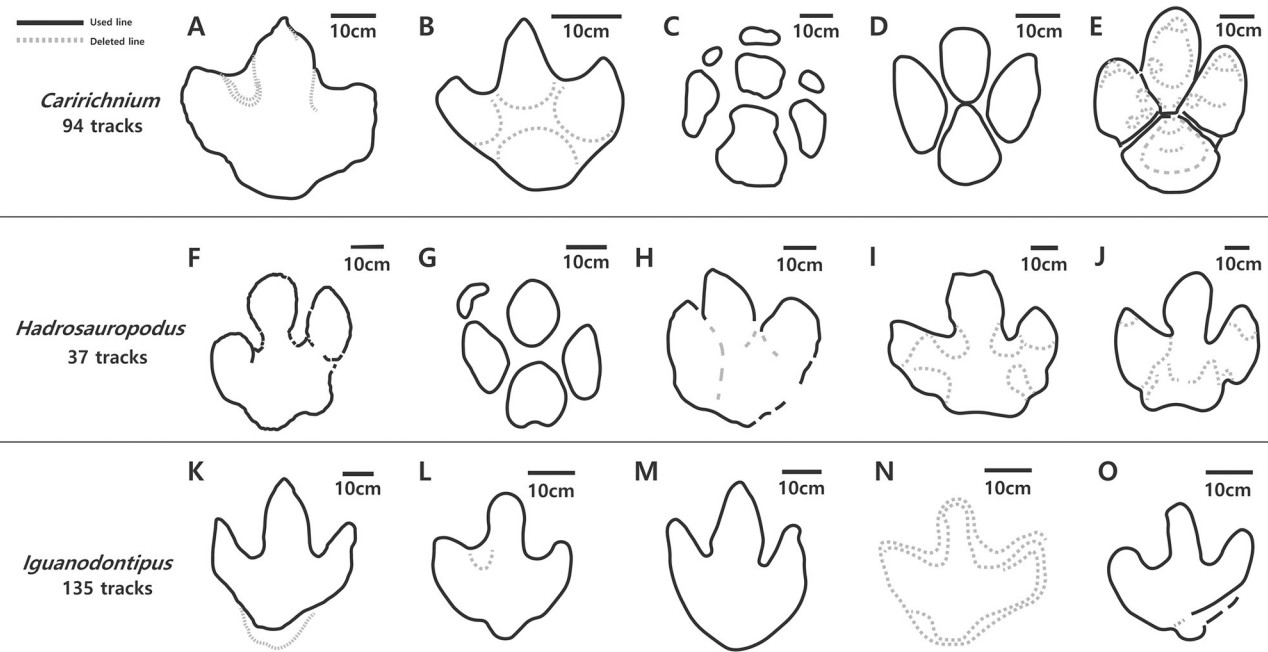

**Fig 3. Examples of redrawn footprint illustrations.** The solid lines represent the outlines of the original footprints included in the train dataset, whereas the dotted lines represent the excluded outlines. A, *Amblydactylus kortmeyeri* [42]; B, *Iguanodontipus billsarjeanti* [8]; C, *Caririchnium* [41]; D, *Caririchnium lotus* [44]; E, *Caririchnium magnificum* [29]; F, *Caririchnium* [47]; G, *Caririchnium kyoungsookimi* [50]; H, *Caririchnium* [29]; I–J, *Hadrosauropodus langstoni* [2]; K, *Iguanodontipus burreyi* [53]; L, *Iguanodontipus* [52]; M, *Iguanodontipus* [6]; N–O, *Iguanodontipus* [52]. The scale bar is 10 cm.

manus (Fig 3). The line thickness of the pes outline did not considerably affect the performance of the model when the features were recognized. The footprint image was placed on a white square background with the track length occupying at least 66% of the background length. In this study, we used a 500×500 pixel white background with a line thickness of 9 pt. Finally, the center of the footprint was placed at the center of the background, with the middle toe pointing upward. Thus, we executed a robust feature extraction operation on the input footprint images of large ornithopod dinosaurs.

## Results and discussion

We developed a machine learning model to classify the ichnogenera of large ornithopod dinosaur footprint illustrations according to the ichnotaxon criteria presented by Díaz-Martínez et al. [3]. The machine learning model was trained for 162 epochs using 274 data images, excluding the horizontally flipped images. The training procedures were stopped after 100 more epochs, after which the validation loss function did not improve (Fig 4). The accuracy and loss functions of the machine-learning model tended to saturate at relatively early epochs, indicating that the model had already learned all it could from the given training data. The final accuracy was 96.36% and the validation accuracy was 92.59%. The trained machine-learning model provided a list of classification probabilities for a given footprint image with respect to the labels *Iguanodontipus*, *Caririchnium*, and *Hadrosauropodus*.

In the following section, we demonstrate the performance of the machine learning model with footprint illustrations that were not included in the training dataset (Fig 5).

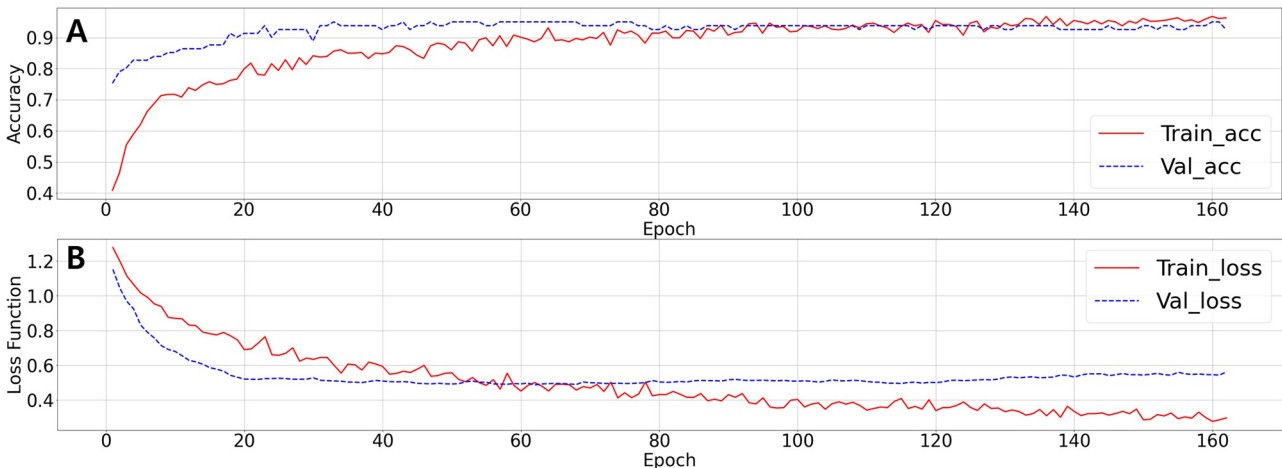

**Fig 4. Changes of accuracy (A) and loss (B) functions of the machine learning model, with the training (red solid) and validation (blue dashed) datasets.** In both trends, the model becomes stable at relatively early epochs.

## *Caririchnium*—Uiesung, Republic of Korea

The dinosaur footprint tracks at Mancheon-ri, Uisung, and Gyeongsangbuk-do in South Korea were reported to be from *Caririchnium* [16]. Although some *Caririchnium* tracks have manus, they were removed during the preconditioning of the image data. The machine learning model classified it as *Caririchnium* with a probability of 97.32% (Fig 5A).

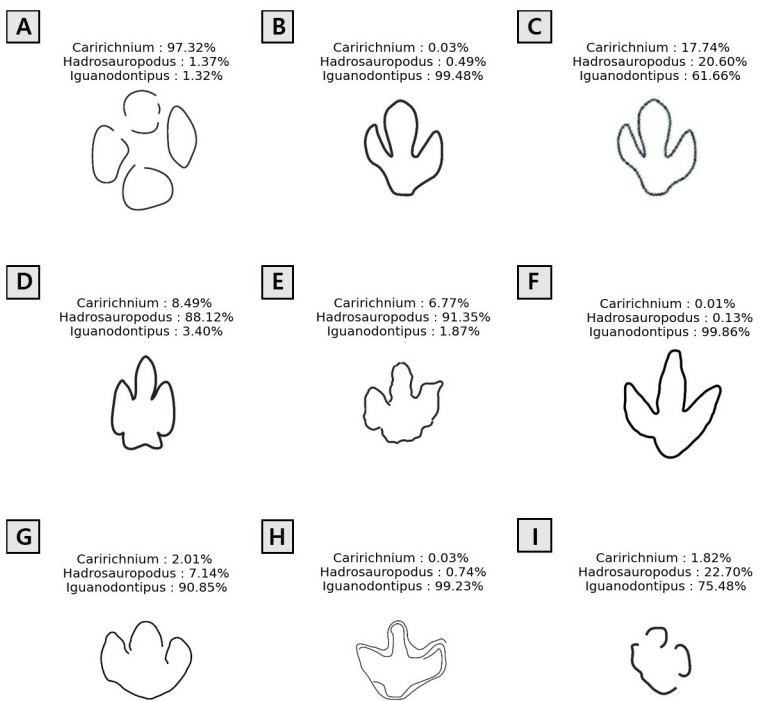

**Fig 5. Test result of large ornithopod tracks classification model.** A, *Caririchnium* from Uiesung, Republic of Korea [16]; B, High quality illustration of *Caririchnium* from Gunbuk, Republic of Korea [26]; C, Low quality illustration of *Caririchnium* from Gunbuk, Republic of Korea [26]; D, *Hadrosauropodus* from La Llau de la Costa, Spain [51]; E, *Hadrosauropodus* from Tetori, Japan [55]; F, *Therangospodus oncalensis* from Tierras Amigas, Spain [13]; G, *Ornithopodichnus* from Sichuan, China [58]; H, *Iguanodontipus* track from Soria, Spain [52] (the footprint is illustrated with multiple lines); I, Ornithopod footprint from Qijiang, China [10].

### *Caririchnium*—Gunbuk, Republic of Korea

The dinosaur footprint tracks at Gunbuk-myeon, Gyeongsangbuk-do, South Korea, have been reported to be from *Caririchnium* [26]. In this example, the original image from the given literature had a low-quality footprint image. The machine learning model classified the footprint as *Iguanodontipus* with a probability of 61.66%, by estimating the lowest classification probability of 17.74% for the *Caririchnium* label (Fig 5C). However, when we enhanced the input image, as shown in Fig 5B, the model classified the footprint as *Iguanodontipus* with a probability of 99.48% and the lowest probability of 0.03% for *Caririchnium*. This discrepancy between the literature [26] and the prediction might result from the input image showing an elongated and narrow digit impression of *Iguanodontipus*, rather than a short, wide digit impression of *Caririchnium*. This result implies the machine learning model can be used as an objective means to validate the classification.

### *Hadrosauropodus*—La Llau de la Costa locality, Spain

The dinosaur footprints found in the Tremp Formation, Spain, were reported as *Hadrosauropodus* [51]. Among the reported tracks, sample MCD-5142, identified in La Llau de la Costa, was used (Fig 5D). The model classified the given input footprint image as *Hadrosauropodus* with a probability of 88.12%.

### *Hadrosauropodus*—Tetori, Japan

The dinosaur footprint (TGUSE-DT1007) found at Tetori, Japan, was reported as being from *Caririchnium* because its digits became slightly narrower distally but remained blunt [55]. However, the machine learning model classified the given footprint (Fig 5E) as *Hadrosauropodus* with a 91.35% probability. Because the given footprint has a bilobed heel, the model classified the input image as *Hadrosauropodus* rather than *Caririchnium*.

### *Iguanodontipus*—Tierras Amigas, Spain

The footprints of *Therangospodus oncalensis* in Tierras Amigas, Spain, were initially reported as theropods and then reinterpreted as the ornithopod *Iguanodontipus oncalensis* [13]. We used samples identified at the Salgar de Sillas site (Fig 5F). The model classified the input footprint as *Iguanodontipus* with a probability of 99.86%.

### *Ornithopodichnus*—Sichuan, China

The footprints found in Sichuan, China, were initially classified as *Ornithopodichnus* [58] and were later invalidated by Díaz-Martínez et al. [3]. The machine learning model showed that the given footprint (Fig 5G) could be classified as *Iguanodontipus* with a 90.85% probability.

### *Iguanodontipus*—Las Cuestas I, Spain

The ornithopod dinosaur footprints found at Las Cuestas I, Spain, were reported as being from *Iguanodontipus* [52], and the initial classification was reconfirmed by Díaz-Martínez et al. [3]. In this example, the footprint illustration is composed of multiple lines on trackway STC-1 (Fig 5H). The model classified the input footprint as *Iguanodontipus* with a probability of 99.23%.

## Ornithopod footprint—Qijiang, China

Numerous ornithopod footprints have been reported for the *Caririchnium lotus* fossil site in Qijiang, China, which is enclosed within the Cretaceous Jiaguan Formation [10]. Although the distribution of fossil tracks was mapped for the entire site, some footprints were poorly preserved; hence, the illustrations did not show the distinctive morphological characteristics of the considered footprints, as shown in Fig 5I. Although the footprint was taken from the trackway of the *Caririchnium lotus*, the model classified it as *Iguanodontipus* with a probability of 75.48%. This indicates that the classification probability tends to be inaccurate for relatively incomplete outlines of fossil tracks. In particular, the corresponding reference [10] only provides a track site map, not the overall image of the fossil tracks.

## Machine learning model for footprints classification

This large ornithological dinosaur footprint illustration classification model showed over 92% learning and validation accuracy. Machine learning models are expected to make researchers' judgments regarding ornithopod dinosaur footprints more reasonable and sophisticated. The machine-learning model will likely prevent erroneous naming of the large ornithopod dinosaur footprint ichnotaxa and act as a precaution against the oversplitting problem.

From the testing results listed above, we found that the machine learning model developed in this study was properly trained to classify any given footprint illustration data for large ornithopod dinosaurs. For example, although the *Hadrosauropodus* footprints reported in Tetori, Japan [55] were not included in the training data, the model classified the given data with a high probability to this ichnotaxon. However, the *Caririchnium* footprint reported in Gunbuk, Republic of Korea [26] showed a substantially different classification result (Iguanodontipus) between the model and literature. As the Gunbuk footprint has an elongated and narrow digit impression (Fig 5B), the classification results appear to be consistent with the standards for large ornithopod ichnotaxa [3].

However, the performance of our machine-learning model is inherently dependent on the quality of the footprint illustration data. For example, the classification probabilities for the footprint data shown in Fig 5B and 5C were estimated differently because of the image quality, although the classification was identical for both cases. Furthermore, the poorly preserved track illustration (Fig 5I) shows results similar to the classification probabilities of the low-quality images (Fig 5C). These discrepancies may be because the model was trained primarily with high-quality data. However, the model was also capable of classifying footprint illustrations with multiple outlines (Fig 5H). This indicates that excessively thick or thin footprint outlines would reduce the classification performance of this model because such lines can hinder feature extraction in the model. Therefore, the model performs better with well-preserved footprints.

Because the machine-learning model developed in this study is a typical supervised learning model, it is impossible to introduce a new label or class. For example, the *Ornithopodichnus* reported in Sichuan, China [56], was excluded from the ornithopod ichnogenera by Díaz-Martínez et al. [3]. Therefore, the labels used to train the model did not include *Ornithopodichnus* [56, 57]. If the reference ichnogenus [3] needs to be revised or a new ichnogenus introduced, the model must be retrained with the footprint datasets available at that time.

In addition, employing machine learning in fossil studies can minimize subjectivity and reduce information loss when delineating fossil footprints. Lallensack et al. [58] were the first to introduce a machine learning technique to differentiate between tridactyl tracks formed by theropods and ornithopods. They trained a deep convolutional neural network (DCNN) using fossil silhouettes. Since our study emphasizes the classification of ornithopod

ichnotaxa as established by Díaz-Martínez et al. [3], it's crucial for researchers to meticulously choose the ornithopod footprints before deploying the model. However, both studies are inherently limited by the information available from interpretive fossil outlines. This constraint can be addressed by broadening the database to encompass three-dimensional fossil morphology [58].

In summary, we developed a machine-learning model to classify large ornithological dinosaur footprints with an overall learning and validation accuracy of 92%. Although this study used illustrations rather than photos or 3D data, the machine learning model will likely prevent misnaming of large ornithopod dinosaur footprint ichnotaxa, thereby safeguarding against the oversplitting problem.

## Acknowledgments

We appreciate Academic Editor Dawid Surmik for his support of this work and the insightful comments and constructive reviews provided by Anthony Romilio and James O. Farlow.

## Author Contributions

**Conceptualization:** Seung-Sep Kim.

**Data curation:** Yeoncheol Ha.

**Formal analysis:** Yeoncheol Ha.

**Funding acquisition:** Seung-Sep Kim.

**Investigation:** Yeoncheol Ha.

**Methodology:** Seung-Sep Kim.

**Project administration:** Seung-Sep Kim.

**Validation:** Seung-Sep Kim.

**Writing – original draft:** Yeoncheol Ha.

**Writing – review & editing:** Seung-Sep Kim.

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
