## [Decision Letter · Decision Letter 0]

14 Jul 2023

PONE-D-23-08114Classification of large ornithopod dinosaur footprints using Xception transfer learningPLOS ONE

Dear Dr. Kim,

Thank you for submitting your manuscript to PLOS ONE. After careful consideration, we feel that it has merit but does not fully meet PLOS ONE’s publication criteria as it currently stands. Therefore, we invite you to submit a revised version of the manuscript that addresses the points raised during the review process.

We look forward to receiving your revised manuscript.

Kind regards,

Dawid Surmik, PhD

Academic Editor

PLOS ONE

Journal Requirements:

“This study was supported by the National Research Foundation of Korea (NRF-2021R1A4A5026233 and NRF-2021R1A2C1012030), awarded to S.-S.K. The funder had no role in study design, data collection and analysis, decision to publish, or preparation of the manuscript.”

“This study was supported by the National Research Foundation of Korea (NRF-2021R1A4A5026233 and NRF-2021R1A2C1012030).”

“This study was supported by the National Research Foundation of Korea (NRF-2021R1A4A5026233 and NRF-2021R1A2C1012030), awarded to S.-S.K. The funder had no role in study design, data collection and analysis, decision to publish, or preparation of the manuscript.”

Additional Editor Comments (if provided):

Dear Authors,

Thank you for submitting your work for publication in PLoS One.

I kindly ask you to refer to the reviewers' suggestions in the minor revision.

All the best, Dawid Surmik

Reviewers' comments:

Reviewer's Responses to Questions

**Comments to the Author**

1. Is the manuscript technically sound, and do the data support the conclusions?

Reviewer #1: Yes

Reviewer #2: Yes

2. Has the statistical analysis been performed appropriately and rigorously? 

Reviewer #1: Yes

Reviewer #2: Yes

3. Have the authors made all data underlying the findings in their manuscript fully available?

Reviewer #1: Yes

Reviewer #2: Yes

4. Is the manuscript presented in an intelligible fashion and written in standard English?

Reviewer #1: Yes

Reviewer #2: Yes

5. Review Comments to the Author

Reviewer #1: Reviewer Summary:

The paper titled "Classification of Large Ornithopod Dinosaur Footprints Using Xception Transfer Learning" introduces a novel approach to tackle oversplitting challenges in the classification of large ornithopod dinosaur footprints. The authors adopt a convolutional neural network-based Xception transfer learning technique to automate the classification process and demonstrate that remarkable accuracy can be attained even with a relatively small supervised image training set (less than 300 images).

Based on my evaluation, I firmly believe that this paper deserves publication (once revisions have been completed) primarily due to its implementation of a machine learning model, which represents a positive and promising step forward in the field of dinosaur ichnological studies. The utilization of advanced computational techniques holds substantial potential in enhancing the accuracy and efficiency of footprint classification, thus contributing to the advancement of this scientific domain.

My main concern revolves around the central question addressed by the authors in the current version of the paper: the oversplitting of large ornithopod dinosaur footprints. It is important to clarify that the primary objective of the paper is not to comprehensively resolve the oversplitting issue, but rather to validate the findings of Díaz-Martínez et al. (2015). This clarification does not diminish the merit of the paper; however, it explains the reliance on the findings of Díaz-Martínez et al. (2015) and the inclusion of only three ichnogenera (Caririchnium, Hadrosauropodus, and Iguanodontipus) in the training set. Consequently, the evaluation of test images is naturally limited to these three ichnotaxa. This focus does not diminish the merit of the paper but rather underscores the strength of their method in verifying ichnotaxonic assignments, albeit based on outlines.

Additionally, in the abstract, the authors claim that their approach represents the first application of machine-learning techniques at the academic level for large ornithopod dinosaur footprints. However, this claim is incorrect. Notably, Lallensack et al. (2022), which is cited in the authors' paper, is actually the first application of machine-learning techniques at the academic level for large ornithopod dinosaur footprints. Lallensack et al.'s focus was on using machine learning to distinguish between tridactyl tracks formed by theropods and ornithopods (not sauropods as stated on page 15). To address this discrepancy, I have two recommendations: 1) Introduce the Lallensack et al. study in the revised manuscript's introduction to highlight the novelty and significance of the method employed. Moreover, clarify the distinctions between Lallensack et al.'s work and the current study, as Lallensack et al. specifically addressed the problem of misidentifying ornithopod tracks as theropodan and vice versa. 2) State that the present study represents the first application of machine-learning techniques at the academic level for classifying large ornithopod dinosaur footprints. It is important to clarify that this paper does not provide a comprehensive evaluation (there are large-bodied ornithopod tracks that were not included in Díaz-Martínez et al study) but rather follows the established work of Díaz-Martínez et al. (2015), which is widely accepted in the field.

Other comments/suggests accompany pdf

Reviewer #2: This is an interesting study of an emerging approach to comparing shapes of tridactyl footprints. While I have some reservations about how well outlines of footprints capture their overall shape, I nonetheless think your approach is worth pursuing.

6. PLOS authors have the option to publish the peer review history of their article (what does this mean?). If published, this will include your full peer review and any attached files.

Reviewer #1: **Yes: **Anthony Romilio

Reviewer #2: **Yes: **James O. Farlow

---

## [Author Response · Author response to Decision Letter 0]

2 Oct 2023

Our detailed responses to the reviews are listed in the rebuttal letter, uploaded to the system.

---

## [Editor Report · Decision Letter 1]

4 Oct 2023

Classification of large ornithopod dinosaur footprints using Xception transfer learning

PONE-D-23-08114R1

Dear Dr. Kim,

We’re pleased to inform you that your manuscript has been judged scientifically suitable for publication and will be formally accepted for publication once it meets all outstanding technical requirements.

Kind regards,

Dawid Surmik, PhD

Academic Editor

PLOS ONE
---

## [Editor Report · Acceptance letter]

9 Oct 2023

PONE-D-23-08114R1 

Classification of large ornithopod dinosaur footprints using Xception transfer learning 

Dear Dr. Kim:

I'm pleased to inform you that your manuscript has been deemed suitable for publication in PLOS ONE. Congratulations! Your manuscript is now with our production department. 

Kind regards, 

on behalf of

Dr. Dawid Surmik 

Academic Editor

PLOS ONE